# Pilot Proteomic Analysis of Urinary Extracellular Vesicles Supports the “Toxic Urine Hypothesis” as a Vicious Cycle in Refractory IC/BPS Pathogenesis

**DOI:** 10.3390/ijms27010130

**Published:** 2025-12-22

**Authors:** Man-Jung Hung, Evelyn Yang, Tsung-Ho Ying, Peng-Ju Chien, Ying-Ting Huang, Wen-Wei Chang

**Affiliations:** 1Department of Obstetrics and Gynecology, Chung Shan Medical University Hospital, Taichung 402306, Taiwan; adiposehung@gmail.com (M.-J.H.); evelynyang0804@gmail.com (E.Y.); ying.steve@gmail.com (T.-H.Y.); tita65@ms33.hinet.net (Y.-T.H.); 2Department of Obstetrics and Gynecology, School of Medicine, College of Medicine, Chung Shan Medical University, Taichung 402306, Taiwan; 3Department of Biomedical Sciences, Chung Shan Medical University, Taichung 402306, Taiwan; 4Department of Medical Research, Chung Shan Medical University Hospital, Taichung 402306, Taiwan

**Keywords:** interstitial cystitis/bladder pain syndrome (IC/BPS), urinary extracellular vesicles, proteomics, toxic urine hypothesis, refractory IC/BPS

## Abstract

Despite treatments such as pentosan polysulfate, hyaluronic acid, botulinum toxin A, and platelet-rich plasma, many interstitial cystitis/bladder pain syndrome (IC/BPS) patients experience persistent symptoms. Urinary extracellular vesicles (uEVs) carry molecular cargo reflecting disease pathophysiology, yet their proteomic profiles in treated IC/BPS remain unexplored. This pilot study examined uEV proteomics in refractory IC/BPS cases to test the “Toxic Urine Hypothesis”—a vicious cycle, whereby urothelial dysfunction enables EV-mediated toxin penetration, triggering inflammation that further impairs the bladder barrier. Urinary EVs were isolated from six female IC/BPS patients on active treatments and four healthy female controls. Mass spectrometry-based proteomics identified differential protein expressions, followed by pathway enrichment analysis and functional validation using NF-κB reporter assays in HEK293T cells and Western blot in primary human bladder epithelial cells. IC/BPS EVs exhibited 31 upregulated proteins (including HPGD, KRT8, HSPA4, 14-3-3 family members) and 19 downregulated proteins (including neutrophil granule proteins MPO and ELANE), indicating suppressed acute neutrophil inflammation but enriched homeostatic, metabolic, and regenerative pathways. Patient EVs induced significantly higher NF-κB activation than in the controls, with upregulated 14-3-3ζ and phosphorylated NF-κB p65 in bladder epithelial cells. These findings support the “Toxic Urine Hypothesis”, revealing persistent NF-κB-mediated chronic epithelial stress despite suppressed acute inflammation in treated IC/BPS patients, suggesting that therapies targeting inflammation and regeneration may help break this vicious cycle.

## 1. Introduction

Interstitial cystitis/bladder pain syndrome (IC/BPS) is a chronic condition characterized by bladder pain and lower urinary tract symptoms, with diagnosis and subclassification primarily based on characteristic cystoscopic and histological findings such as glomerulations and Hunner lesions [1,2]. Despite ongoing research, the etiology and optimal management of IC/BPS remain unclear due to the incomplete understanding of its underlying pathophysiological mechanisms. Urothelial dysfunction, often due to repeated bladder insults, is believed to permit urinary toxins, pathogens, and cationic substances to penetrate the suburothelial layer, resulting in chronic inflammation and neural sensitization [3,4].

Current treatment options include oral pharmacological agents—such as pentosan polysulfate sodium (PPS, the only FDA-approved medication for IC/BPS [5]), anti-inflammatory agents, and analgesics—as well as intravesical instillation therapies using hyaluronic acid (HA), chondroitin sulfate (CS), or dimethyl sulfoxide (DMSO). These interventions aim to restore urothelial integrity and reduce inflammation [6,7,8]. More severe or refractory cases may benefit from interventions such as intravesical botulinum toxin A injection [9] and platelet-rich plasma (PRP) [10], which have both been shown to alleviate refractory symptoms via anti-inflammatory and regenerative effects [11]. Adjunct approaches such as pelvic floor rehabilitation and dietary modification may provide additional symptomatic relief [12,13,14].

Recent research underscores the utility of urinary biomarkers—including MCP-1, RANTES, CXCL-10, IL-7, eotaxin-1, and subtype-associated markers such as BDNF and IL-8—in improving diagnostic accuracy and enabling disease stratification [15,16,17]. The combined elevation of eotaxin and TNF-α has demonstrated utility in distinguishing IC/BPS from other lower urinary tract dysfunctions [18], facilitating personalized treatment strategies.

Extracellular vesicles (EVs), including exosomes and microvesicles, are gaining attention due to their rich cargo—proteins, lipids, and nucleic acids—implicated in cell communication and immune regulation [19,20]. Notably, EVs derived from urine or stem cells have shown promise in modulating bladder inflammation and function, and urinary exosomal markers (e.g., MEG3, miR-9) have demonstrated potential diagnostic and therapeutic significance in IC/BPS [21,22,23]. However, the pathological and diagnostic roles of urinary EV cargo proteins remain insufficiently characterized.

Despite these advances, the molecular mechanisms contributing to refractory symptoms and incomplete therapeutic response in IC/BPS require further clarification. The “Toxic Urine Hypothesis” posits that urinary EVs may mediate persistent toxicity and inflammation, perpetuating chronic inflammation and tissue injury. This pilot case–control study explores proteomic changes in urinary EVs from a small cohort of refractory IC/BPS patients receiving various treatments—including HA, PRP, PPS, or botulinum toxin A—versus healthy controls, by utilizing mass spectrometry-based profiling and functional assays to assess their inflammatory potential. By integrating proteomic and mechanistic analyses, this study aims to identify new biomarkers for treatment monitoring and to assist in the development of more effective interventions for IC/BPS. The data are intended as preliminary hypothesis-generating evidence to guide future larger-scale investigations.

## 2. Results

### 2.1. Patient Characteristics

Six female patients with refractory interstitial cystitis/bladder pain syndrome (IC/BPS), who underwent bladder hydrodistension and injection therapy (BOTOX and/or PRP) between October and November 2024 with failure of response to prior intravesical hyaluronic acid (HA) treatment, were enrolled. Four healthy female volunteers with no urinary disorders (confirmed by history) served as controls. Characteristics, including age, body mass index, disease duration (from cystoscopic diagnosis), Interstitial Cystitis Symptom Index (ICSI) and Problem Index (ICPI) scores, pain visual analog scale (VAS) scores, 24-h urination frequency, average voided volume, functional/anesthetic bladder capacity, and glomerulation grades post-hydrodistension, are summarized in Table 1. Patients exhibited chronic symptoms and cystoscopic findings (e.g., glomerulations) despite ongoing therapy.

### 2.2. Characterization of Urinary Extracellular Vesicles (uEVs)

The NTA showed EV size distributions peaking at 100–150 nm in both healthy controls (Figure 1A) and IC/BPS patients (Figure 1C). TEM confirmed membrane-enclosed cup-shaped vesicles in both groups (Figure 1B,D). Western blot revealed the presence of common EV markers (ALIX, TSG101, CD9) in both in both groups, but not the negative marker calnexin (Figure 1E). The expression levels of the EV markers were similar between the control group and the IC/BPS patients, supporting successful uEV isolation for subsequent proteomic analysis. Quantitative abundance of these common EV markers via mass spectrometry was provided in Appendix A. The mean particle sizes (180–250 nm) and concentrations were also similar between groups (Figure 1F,G).

### 2.3. Proteomic Analysis of uEVs Reveals the Suppression of Neutrophil-Mediated Inflammation in Treated IC Patients

To understand the molecular composition of urinary EVs, a mass spectrometry-based proteomic analysis was conducted. PCA of the top 200 most variable proteins demonstrated clear separation between the two groups, with the first two principal components accounting for 55.6% of the total variance (PC1: 36.2%, PC2: 19.4%) (Figure 2A). This distinct clustering pattern indicates that the proteomic profiles of the uEVs from treated IC patients remain substantially different from those of the control group. The volcano plot visualization revealed numerous proteins meeting the criteria of log2 fold change >1 and *p*-value < 0.05. Several common EV markers were detected, including ALIX, CD9, CD63, HSP70, and TSG101. Of these, only HSP70 increased significantly in IC/BPS patients (Appendix A). A total of 31 proteins were found to be upregulated in uEVs from IC patients, including HPGD (15-Hydroxyprostaglandin dehydrogenase), KRT8 (keratin-8), HSPA4 (Heat Shock Protein Family A (Hsp70) Member 4), and YWHAZ (Tyrosine 3-Monooxygenase/Tryptophan 5-Monooxygenase Activation Protein Zeta or 14-3-3ζ). Meanwhile, 19 proteins were found to be downregulated, including DEFA1 (Defensin Alpha 1), MPO (myeloperoxidase), SERPINB12 (Serpin Family B Member 12), and S100A8 (S100 Calcium Binding Protein A8) (Figure 2B). The hierarchical clustering analysis (Figure 2C) reveals that treated IC patients possess a fundamentally altered urinary EV proteome characterized by two coordinated molecular programs. One is the suppression of neutrophil-mediated inflammation, indicated by the downregulation of neutrophil granule proteins, such as MPO, ELANE (neutrophil elastase), CTSG (cathepsin G), AZU1 (azurocidin), and LTF (lactotransferrin). The other is the activation of cellular homeostatic and regenerative processes, indicated by the upregulation of 14-3-3 protein family, such as YWHAZ, YWHAB, YWHAE, and YWHAG, and the cytoskeletal/structural proteins, such as VCL (vinculin), KRT8, and CAPG (capping protein, gelsolin-like).

### 2.4. Functional Enrichment Analysis Reveals Distinct Biological Pathways Associated with Protein Alterations in IC Patient uEVs

Pathway enrichment (Gene Ontology [GO], Kyoto Encyclopedia of Genes and Genomes [KEGG], Reactome) for upregulated proteins showed cellular homeostasis, structure, and metabolism (Figure 3A). Downregulated proteins were enriched for inflammatory and immune defense pathways (Figure 3B). Gene Set Enrichment Analysis (GSEA) using Hallmark gene sets showed enrichment trends toward mTORC1 signaling, MYC targets, glycolysis, estrogen response, fatty acid metabolism, and PI3K-AKT-mTOR signaling in treated IC/BPS patients, though these did not reach statistical significance after multiple testing correction (Figure 4A). These patterns may suggest potential PRP-mediated metabolic and proliferative signaling during tissue regeneration. The GSEA of GO C5 terms revealed the upregulation of metabolic processes, including nucleotide metabolism, purine-containing compound metabolism, and mitochondrial organization in treated IC/BPS patients (Figure 4B). On the other hand, pathways related to the innate immune defense showed downregulation, including defense responses to Gram-negative bacteria and fungi, antimicrobial humoral responses, and cell killing processes. In addition, external encapsulating structures also exhibited reduced enrichment (Figure 4B). These patterns suggest metabolic activation accompanied by the suppression of innate immune responses in treated IC/BPS patients, potentially reflecting disease remission or therapeutic modulation of inflammatory pathways.

### 2.5. Urinary EVs from IC Patients Enhance NF-κB Activation

Despite treatment (HA, PRP, pentosan polysulfate, or botulinum toxin A), patients had persistent pain and bleeding. Given elevated 14-3-3 proteins and metabolic signatures in uEVs, we tested their pro-inflammatory role via NF-κB reporter assay in HEK293T cells (Figure 5A). Patient uEVs induced significantly higher NF-κB activation than controls (*p* < 0.05; TNF-α as positive control, pGL3-Basic as negative) (Figure 5B). We further treated primary human bladder epithelial cells (HBlEpC) with the isolated uEVs. Due to limited uEV yield from individual samples, this functional validation was performed using uEVs from one healthy donor and one IC/BPS patient. Patient-derived uEVs upregulated 14-3-3ζ and phosphorylated NF-κB p65 (ser536) (Figure 5C). This confirms uEVs’ functional inflammatory effects, potentially driving persistent symptoms via NF-κB-mediated pain, permeability, and treatment resistance.

## 3. Discussion

This pilot proteomic analysis of uEVs from a small cohort of refractory IC/BPS patients revealed a paradoxical signature: upregulated homeostatic and regenerative proteins alongside persistent NF-κB activation. These findings elucidate key mechanisms underlying treatment resistance and identify potential biomarkers and therapeutic targets, supporting the Toxic Urine Hypothesis as a vicious cycle of urothelial dysfunction, EV-mediated inflammation, epithelial damage, and neuronal sensitization.

Upregulation of YWHAZ (14-3-3ζ) and other 14-3-3 family members (YWHAB, YWHAE, YWHAG) was prominent in patient uEVs (Figure 2B,C). While YWHAZ promotes inflammation resolution in rheumatoid arthritis via regulatory T cells and M2 macrophages [24], our data show patient uEVs induce robust NF-κB activation comparable with TNF-α (Figure 5B), suggesting context-dependent pro-inflammatory effects in refractory IC/BPS. This may contribute to persistent symptoms despite therapy, highlighting cell-type-specific roles.

Similarly, elevated 15-hydroxyprostaglandin dehydrogenase (HPGD), a prostaglandin catabolizing enzyme linked to urothelial differentiation and barrier integrity [25], reflects compensatory efforts in IC/BPS. Studies show a >2-fold HPGD increase in differentiating IC/BPS urothelial cells, with impaired PGE2 release [26], indicating insufficient management of prostaglandin dysregulation amid chronic barrier defects.

Several identified proteins suggest mechanisms of resistance that perpetuate this vicious cycle. Upregulated HSPA4 may sustain IL-17-mediated inflammation and prevent apoptotic clearance, fostering steroid resistance [27]. Elevated keratins (KRT8, KRT19) indicate urothelial damage, releasing damage-associated molecular patterns (DAMPs) that amplify NF-κB signaling [28]. Downregulated SERPINB12 compromises protease regulation and barrier integrity [29], akin to inflammatory relapse in eosinophilic esophagitis [30]. Collectively, these uEV cargos act as pro-inflammatory mediators, activating NF-κB in recipient cells (Figure 5C) and sustaining epithelial stress, inflammation, and sensitization.

These findings have key implications for the management of refractory IC/BPS. Upregulated YWHAZ, HSPA4, and keratins (e.g., KRT8), with downregulated SERPINB12, could serve as uEV biomarkers for the early identification of treatment resistance, guiding intensified therapies rather than sequential trials. These proteins also represent potential therapeutic targets. HSPA4 inhibition, SERPINB12 restoration, 14-3-3 modulation, or EV signaling blockade could enhance treatment outcomes by addressing EV-driven chronic stress under the Toxic Urine Hypothesis. Serial uEV monitoring might further provide molecular response metrics to guide therapeutic adjustments. Given the limited sample size, treatment heterogeneity, and absence of longitudinal sampling, the present work should be regarded as a pilot study providing preliminary hypothesis-generating data rather than definitive evidence. Future research should employ larger standardized cohorts with serial sampling, correlate uEV profiles with cystoscopic and histopathological properties, and validate functions in cellular/animal models to confirm biomarkers and targets. Additionally, functional validation of NF-κB activation and 14-3-3ζ upregulation in primary bladder epithelial cells was limited to uEVs from single representative healthy and IC/BPS samples due to constrained EV yields, precluding statistical analysis. Future studies with optimized EV isolation protocols or pooled samples should validate these trends in larger cohorts.

In recent years, the combination of ultrafiltration combination with size-exclusion chromatography (UF-SEC) has emerged as the most frequently employed protocol for uEV isolation in proteomic studies, owing to its superior removal of soluble contaminants such as albumin and uromodulin [31]. In the present pilot study, we intentionally used ultrafiltration alone (with 100 kDa cutoff filters) to maximize EV recovery from the limited urine volumes available from our small clinical cohort (50 mL per subject). This choice preserved sufficient material for deep proteomic profiling; however, as expected, it resulted in the co-isolation of abundant non-EV proteins, including albumin and uromodulin, which were identified in the LC-MS/MS data without differential expression between groups. Western blotting confirmed the absence of cellular contamination (calnexin-negative; Figure 1E) and the presence of common EV markers (e.g., CD9, ALIX, TSG101) in both groups (Figure 1E). In the proteomic data, there were no statistical differences in the abundance of the common EV makers between IC/BPS patients and healthy donors except for HSP70, which was increased in IC/BPS patients (Appendix A). Although more than 600 proteins were still identified in uEV samples from healthy donors or IC/BPS patients, the reduced dynamic range inherent to ultrafiltration-only preparations likely masked lower-abundance candidates. For future validation studies or larger cohorts, adopting UF-SEC or additional orthogonal purification steps (e.g., density-gradient centrifugation or uromodulin-reducing treatments) to achieve higher purity and higher confidence in low-abundance biomarkers should be considered [32].

## 4. Materials and Methods

### 4.1. Study Design and Patient Enrollment

This case–control study was conducted at Chung Shan Medical University Hospital (Taichung, Taiwan) and approved by its Institutional Review Board (approval No. 2024001). All participants provided written informed consent. Female patients with interstitial cystitis/bladder pain syndrome (IC/BPS), diagnosed by clinical symptoms and cystoscopic findings (glomerulations or Hunner lesions), were enrolled if their clinical symptoms persisted for more than 6 months after treatment with intravesical hyaluronic acid (HA) instillation (global response assessment score <+2; range −3 to +3) 24. Eligible patients underwent additional bladder hydrodistension with injection therapy (botulinum toxin A [BOTOX] and/or platelet-rich plasma [PRP]). Exclusion criteria included active urinary tract infection, bladder malignancy, or other pelvic pain syndromes. Healthy female controls were age-matched (±5 years) volunteers with no urinary disorders (confirmed by history and negative Interstitial Cystitis Symptom Index [ICSI]/Problem Index [ICPI] scores). First-morning midstream urine samples (50 mL) were collected pre-procedure for extracellular vesicle (EV) isolation and analysis. Six IC/BPS patients and four controls were enrolled.

### 4.2. Urinary EV (uEV) Isolation

The urine samples underwent differential centrifugation at 4 °C: 300× *g* for 10 min to remove cells, followed by 2000× *g* for 30 min to remove debris. The supernatant (30 mL) was concentrated to approximately 200 µL using Amicon Ultra-15 filters (100 kDa cutoff; MilliporeSigma, Burlington, MA, USA) at 13,000× *g*, washed with 2 mL phosphate-buffered saline (PBS), filtered through a 0.22 µm membrane, and stored at −20 °C.

### 4.3. Nanoparticle Tracking Analysis (NTA)

EV size and concentration were analyzed using NanoSight Pro (Malvern Panalytical Ltd., Malvern, UK) at the Medical and Pharmaceutical Industry Technology Development Center (New Taipei City, Taiwan). The instrument was equipped with an sCMOS camera, syringe-pump-driven flow cell for continuous sample delivery, and a 532 nm laser (maximum power <50 mW) for light-scatter detection. Samples were diluted in particle-free PBS to 20–100 particles/frame. Five 60 s videos were recorded per sample and analyzed with NTA software (version 2.0, Malvern Panalytical Ltd., Malvern, UK) with default settings, including finite track length adjustment and minimum track length enabled.

### 4.4. Transmission Electron Microscopy (TEM)

EV ultrastructure was examined using a JEM-1400 TEM (National Cheng Kung University, Tainan, Taiwan). The EV suspension (10 µL) was adsorbed on formvar-carbon copper grids (400 mesh) for 20 min, negatively stained with 2% uranyl acetate for 5 min, air-dried, and imaged at 80 kV.

### 4.5. Proteomic Analysis

EV proteins were extracted in SDS buffer, separated by SDS-PAGE (1 cm gels), excised, and trypsin-digested overnight at 37 °C. Peptides were analyzed via one-dimensional liquid chromatography–tandem mass spectrometry (LC-MS/MS) using an LTQ-Orbitrap (Thermo Fisher Scientific, Waltham, MA, USA) in data-dependent mode; full MS was acquired in the Orbitrap and MS/MS in the linear ion trap. Data were searched against a protein database using Mascot (Matrix Science, London, UK), with the search parameters set to trypsin digestion (maximum 2 missed cleavages), carbamidomethylation (C, fixed), methionine oxidation (M, variable), and standard mass tolerances. Proteins were identified based on Mascot scores and the detection of at least two unique peptides. Differential expression analysis was performed using label-free quantification, applying specific fold-change and statistical thresholds. Analyses, including identification of differential proteins, principal component analysis (PCA) plot, heatmap generation, and pathway enrichment, were performed with R ggplot2 (R Foundation for Statistical Computing, Vienna, Austria).

### 4.6. Western Blot Analysis

EV proteins were extracted using RIPA buffer, separated (20 µg per lane) on 10% SDS-PAGE, and transferred onto 0.45 µm PVDF membranes (Pall Corporation, Port Washington, NY, USA). Membranes were blocked with 5% nonfat milk in TBS for 1 h at room temperature, incubated with primary antibodies at 4 °C for 16 h, washed with 0.1% Tween-20/TBS, and probed with HRP-conjugated secondary antibodies for 1 h at room temperature. Signals were developed using an ECL substrate (Thermo Fisher Scientific) and visualized with an Amersham Imager 680 (Cytiva, Marlborough, MA, USA). For the detection of cellular proteins, human bladder epithelial cells (HBlEpC; Cell Applications, Inc., San Diego, CA, USA) were treated with uEVs for 48 h and lysed with RIPA buffer, followed by SDS-PAGE and immunoblot analysis using the indicated antibodies. Details of all the antibodies used in this study are provided in Appendix A.

### 4.7. Cell Culture

HEK293T cells were obtained from the American Type Culture Collection (ATCC, Manassas, VA, USA; cat. no. CRL-11268). Primary human bladder epithelial cells (HBlEpC) were purchased from Cell Applications, Inc. (San Diego, CA, USA; cat. no. 938K-05a). HEK293T cells were cultured in Dulbecco’s Modified Eagle Medium (DMEM) supplemented with 10% fetal bovine serum (FBS; HyClone, Logan, UT, USA) and 1× Antibiotic–Antimycotic reagent (Gibco, Thermo Fisher Scientific, Waltham, MA, USA). HBlEpC cells were cultured using the HBlEpC Growth Medium Kit provided by Cell Applications, Inc. (Cat No. 217K-500), according to the manufacturer’s protocol. Cells were maintained at 37 °C in a humidified incubator with 5% CO_2_. HEK293T cells were authenticated by short tandem repeat (STR) profiling performed at the Center for Genomic Medicine (National Cheng Kung University, Tainan, Taiwan). As HBlEpC are primary cells, STR profiling is not applicable. Both cell types were confirmed negative for mycoplasma contamination using the MycoAlert™ Mycoplasma Detection Kit (Lonza, Basel, Switzerland) prior to experiments.

### 4.8. Luciferase Reporter Assay

HEK293T cells (1 × 10^5^/well) were seeded in 12-well plates in DMEM (10% FBS, 1% penicillin/streptomycin; 37 °C, 5% CO_2_). After 16 h, cells were co-transfected with pGL3-NF-κB firefly luciferase reporter (NovoPro Bioscience Inc., Shanghai, China) and pRL-TK Renilla control using TransIT (Mirus Bio LLC, Madison, WI, USA). At 32 h post-transfection, cells were treated with EVs (50 µg/mL) or TNF-α (10 ng/mL, positive control) for 24 h; pGL3-Basic served as the negative control. Dual-luciferase activity was measured using the Promega system (Promega Corp., Madison, WI, USA) and detected with a BioTek microplate reader (BioTek Instruments, Inc., Winooski, VT, USA), with NF-κB activity normalized to Renilla.

### 4.9. Statistical Analysis

Quantitative data were expressed as the mean ± standard error of the mean (SEM). Differences between the two groups were assessed using an unpaired *t*-test. For comparisons between multiple groups (more than two), repeated-measures ANOVA followed by Tukey–Kramer post hoc tests were applied to identify significant differences between specific groups. A *p*-value of less than 0.05 was considered statistically significant.

## 5. Conclusions

This pilot study provides an initial comprehensive proteomic characterization of urinary EVs from IC/BPS patients undergoing active treatment. The findings reveal that, despite the suppression of acute neutrophil-mediated inflammation, these patients exhibit persistent NF-κB activating capacity and a complex molecular signature indicative of chronic epithelial stress, differentiation defects, and treatment resistance. These findings support the Toxic Urine Hypothesis, wherein urinary EVs act as carriers of persistent inflammatory mediators, perpetuating a vicious cycle of urothelial dysfunction, toxin penetration, inflammation, and further barrier impairment that contributes to refractory pathogenesis. The identification and functional validation of specific protein biomarkers provide a molecular framework for understanding treatment-resistant IC/BPS and lay a foundation for the development of next-generation diagnostic tools and targeted therapeutic strategies. Overall, the results of this pilot study emphasize that IC/BPS is not only an inflammatory condition but also a multifaceted disorder characterized by coordinated defects in epithelial differentiation, barrier function, prostaglandin homeostasis, and intercellular communication. Effective long-term management necessitates comprehensive therapeutic approaches that simultaneously target inflammation and restore epithelial regeneration to break the vicious cycle and achieve sustained symptom relief and disease remission. Future studies with larger well-characterized cohorts are needed to validate these candidate biomarkers and mechanisms and to translate them into clinically actionable tools.

## Figures and Tables

**Figure 1 ijms-27-00130-f001:**
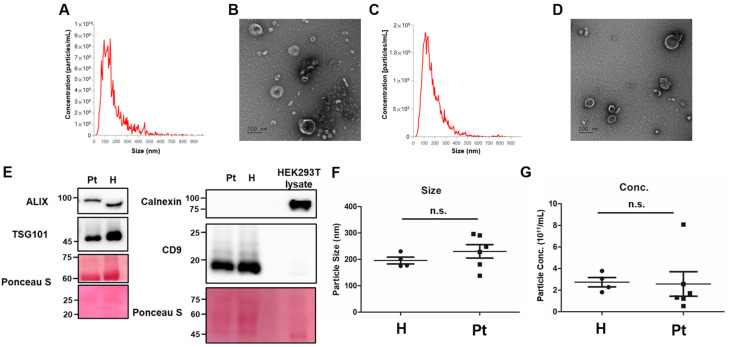
Characterization of urinary extracellular vesicles (uEVs). uEVs were isolated via ultrafiltration from healthy controls and IC/BPS patients. Nanoparticle tracking analysis (NTA) shows the size distribution of EVs from (**A**) healthy controls and (**C**) IC/BPS patients. Transmission electron microscopy (TEM) images show EV morphologies from (**B**) healthy controls and (**D**) IC patients. Scale bars: 200 nm. (**E**) Western blot analysis of common EV markers (ALIX, TSG101, CD9) and the negative marker of calnexin comparing healthy controls (H) and IC/BPS patients (Pt); Ponceau S staining serves as the loading control. Cell lysate of HEK 293T cells was used as a control for calnexin detection. Quantitative proteomic abundance of these common EV markers is shown in Appendix A. The mean particle size (**F**) and particle concentration (**G**) across groups. Horizontal lines represent the mean ± SEM (*n* = 4 in the control group, *n* = 6 in the group of IC/BPS patients). n.s., not significance.

**Figure 2 ijms-27-00130-f002:**
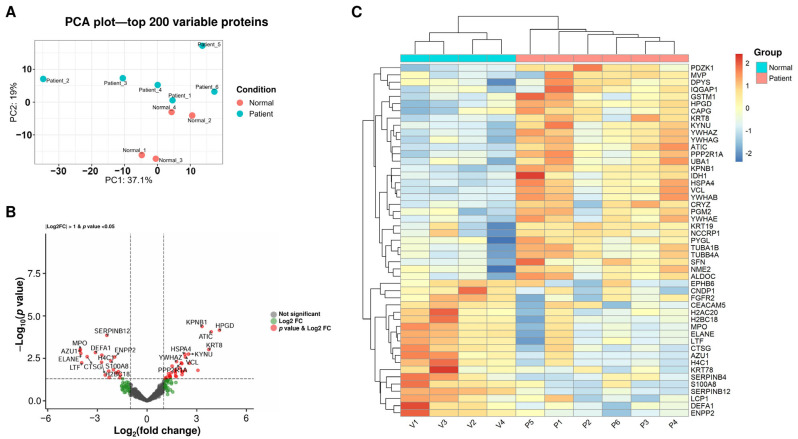
Proteomic profiling of urinary extracellular vesicles distinguishes treated interstitial cystitis patients from healthy controls. (**A**) Principal component analysis (PCA) plot of the top 200 most variable proteins. Samples are plotted based on the first two principal components (PC1: 36.2% variance, PC2: 19.4% variance). Control group samples (Normal, *n* = 4) are shown in red, and treated interstitial cystitis patient samples (Patient, *n* = 6) are shown in light blue. (**B**) Volcano plot of the differential expressed proteins. The x-axis shows the log2 fold change, and the y-axis shows the -log_10_(*p*-value). Proteins with |log_2_FC| > 1 and *p* < 0.05 are shown in red (significantly up- or downregulated). Gray dots represent non-significant proteins. Key proteins are labeled. (**C**) Hierarchical clustering heatmap of differentially expressed proteins. Rows represent individual proteins, and columns represent individual samples. Sample annotations are color-coded at the top: cyan indicates healthy controls (Normal, *n* = 4: V1, V3, V2, V4), and coral/orange indicates treated IC patients (Patient, *n* = 6: P1, P2, P3, P4, P5, P6). Protein expression levels are shown as z-score normalized values, with red indicating higher expression, blue indicating lower expression, and white indicating average expression. The dendrogram on the left shows the hierarchical clustering of proteins based on expression similarity.

**Figure 3 ijms-27-00130-f003:**
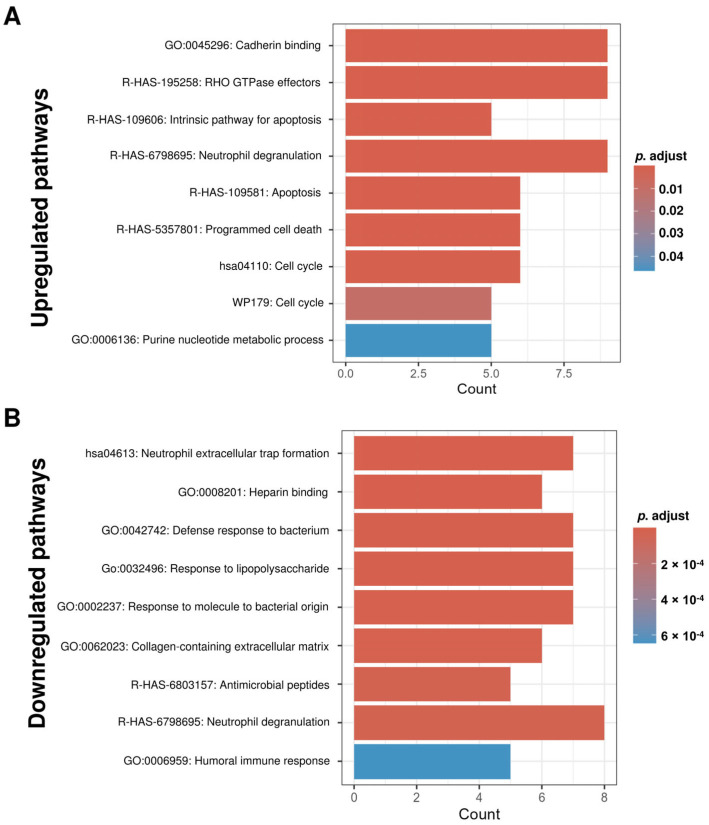
Functional pathway enrichment analysis reveals distinct biological processes associated with differentially expressed proteins in uEVs from treated IC patients. The top enriched Gene Ontology (GO), Kyoto Encyclopedia of Genes and Genomes (KEGG), and Reactome pathways for proteins with higher expression in IC patient uEVs compared with healthy controls are shown in bar plots, with the enriched pathways in the upregulated (**A**) and downregulated (**B**) proteins. Bar colors represent the adjusted *p*-value (*p*. adjust) based on the color scale on the right.

**Figure 4 ijms-27-00130-f004:**
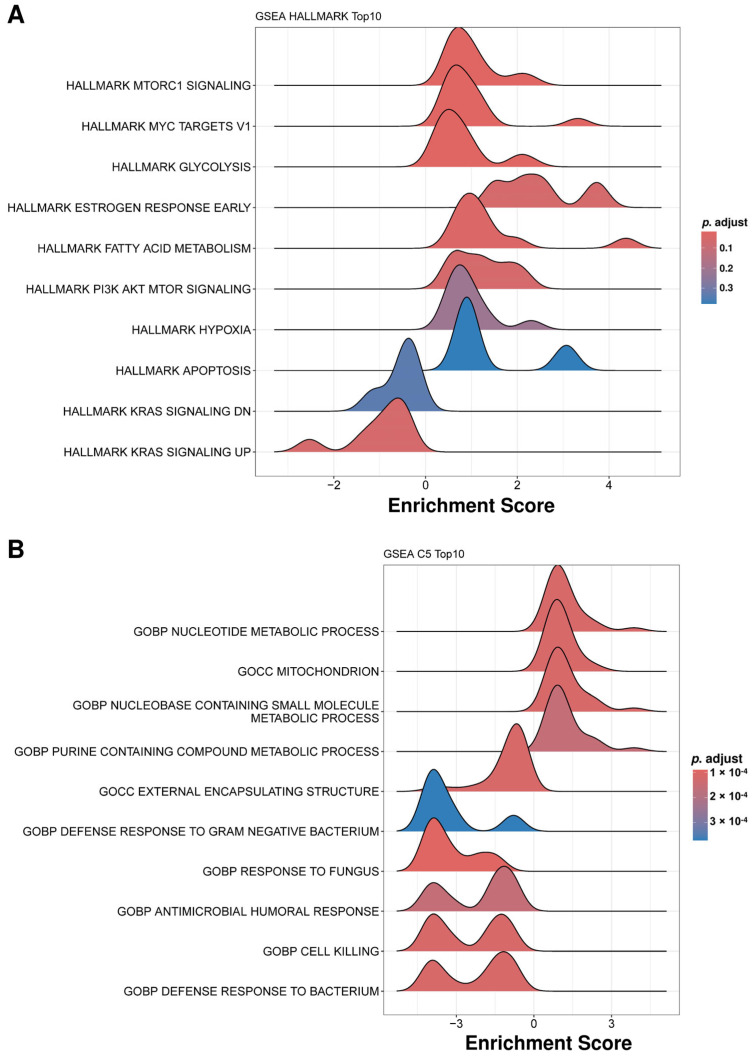
Gene Set Enrichment Analysis (GSEA) reveals metabolic reprogramming and altered immune responses in uEV from treated IC patients. Ridge plots showing the distribution of enrichment scores for the top 10 most enriched Hallmark gene sets (**A**) and C5 gene sets (**B**). Each ridge plot represents a different pathway, with the x-axis showing the enrichment score and the y-axis listing pathway names. The shape and position of the distribution indicate the degree and direction of enrichment. Colors represent the adjusted *p*-value (*p*. adjust).

**Figure 5 ijms-27-00130-f005:**
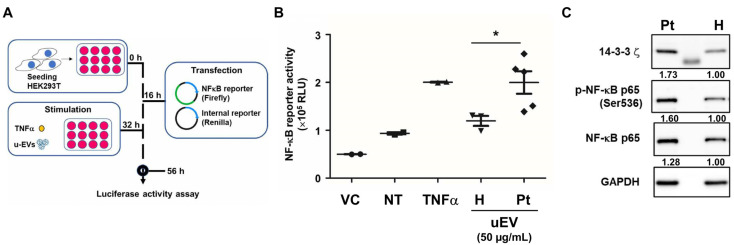
Urinary extracellular vesicles from IC/BPS patients enhance NF-κB activity in HEK293T reporter cells and human bladder epithelial cells. (**A**) Schematic representation of the dual-luciferase reporter assay workflow. HEK293T cells were seeded at 0 h, transfected with NF-κB reporter (Firefly luciferase) and internal control (Renilla luciferase) plasmids at 16 h, stimulated with uEVs (50 µg/mL) or TNFα (10 ng/mL) at 32 h, and assessed for luciferase activity at 56 h. pGL3-Basic vector lacking NF-κB binding sites served as a negative control. (**B**) NF-κB activity represented as normalized Firefly/Renilla luciferase ratios. Data are presented as the mean ± SEM. Statistical significance was determined between the indicated groups: *, *p* < 0.05. (**C**) Here, 50 µg/mL of uEVs from a healthy donor (H) or an IC patient (Pt) was used to treat human bladder epithelial cells (HBlEpC) for 48 h. The protein expressions of 14-3-3ζ, p-NF-κB p65 at serine536, and total NF-κB p65 were determined via Western blot. GAPDH was used as an internal protein loading control. The numbers below the blots indicate expression levels relative to H uEV–treated cells after normalization to GAPDH.

**Table 1 ijms-27-00130-t001:** Clinical and functional characteristics of IC/BPS patients.

IC Patients	Age (years)	BMI (kg/m^2^)	Duration (months)	ESSIC Typing ^1^	Glomerulation Initial/Current ^1^		Cystoscopic Capacity (mL)	Urinary Frequency	VV ^2^ (mL)	FBC ^2^ (mL)	ICSI ^3^ (0–20)	ICPI ^3^ (0–16)	Pain-VAS ^3^	Additional Tx ^3^
1	61	21.9	26	2	3	3	650	8.0 ± 1.0	175.4 ± 10.1	280	10	12	6	PRP
2	36	18.4	97	2	3	3	700	16.7 ± 0.6	51.6 ± 3.6	110	20	16	10	BOTOX, PRP
3	49	21.6	86	2	2	1	600	11.0 ± 1.0	134.0 ± 10.6	300	10	9	2	BOTOX, PRP
4	34	19.4	76	2	2	1	550	9.3 ± 0.6	123.4 ± 9.6	210	4	4	2	PRP
5	47	20.9	28	2	2	2	800	14.7 ± 2.3	57.0 ± 4.9	220	16	15	10	PRP
6	49	21.6	113	2	2	1	850	11.3 ± 2.1	106.7 ± 12.5	260	5	4	4	BOTOX, PRP
Mean ± SD	46.0 ± 9.9	20.6 ± 1.4	71.0 ± 36.2				691.7 ± 115.8	11.8 ± 3.3	108.0 ± 45.2	230.0 ± 68.1	10.8 ± 6.2	10.0 ± 5.3	5.7 ± 3.7	

^1^ Data are shown as medians. ^2^ VV (voided volume) and FBC (functional bladder capacity) were derived from a three-day voiding diary, representing the average and maximum voided volumes, respectively. ^3^ ICSI (Interstitial Cystitis Symptom Index, 0–20), ICPI (Interstitial Cystitis Problem Index, 0–16), and Pain-VAS (Pain Visual Analog Scale, 0–10) were used to evaluate the symptom severity and pain intensity. ^3^ Tx, treatment; PRP, platelet-rich plasma; BOTOX, botulinum toxin A.

## Data Availability

The raw data supporting the conclusions of this article will be made available by the authors upon request. The data are not publicly available due to ethical and privacy concerns involving human subjects.

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
