# Peer review of "Pilot Proteomic Analysis of Urinary Extracellular Vesicles Supports the “Toxic Urine Hypothesis” as a Vicious Cycle in Refractory IC/BPS Pathogenesis"

_ijms, 2025, doi:10.3390/ijms27010130_

Round 1
Reviewer 1 Report
Comments and Suggestions for Authors
Dear Authors,
I enjoyed reading your paper entitled “Proteomic Analysis of Urinary Extracellular Vesicles Supports 2 the “Toxic Urine Hypothesis” as a Vicious Cycle in Refractory 3 IC/BPS Pathogenesis” because I am convinced of the important role of urinary extracellular vesicles (uEV) in pathophysiological processes and their usefulness as an accessible source of biomarkers.
However, there are two main issues that need attention:
- As you yourselves admit, the low number of patients in the analysis make the study weak. I recommend emphasizing throughout the text that this is a pilot study with preliminary data.
- A major issue concerns the choice of uEV isolation method, especially since the study is based on proteomics. Ultrafiltration is not suitable for proteomic analysis, since uEV purity is very low. In my experience, by ultrafiltration you can isolate uEV highly contaminated by albumin and Tamm Horsfall Protein/Uromodulin (UMOD). This aspect is not taken into consideration in the paper, nor is there any description of treatments that serve to reduce the influence of UMOD, which unfortunately has always complicated the proteomic analysis of uEV. Recent papers on uEVs that use ultrafiltration always associate it with Size-Exclusion Chromatography (SEC). Please comment on this question and explain the reason for choosing ultrafiltration. Moreover, it would be advisable to improve the characterization of EVs (Fig. 1E), showing the enrichment of markers in the vesicular fraction in comparison with at least total urinary proteins. Furthermore, since HSP70 is among the differential proteins, I would not use it as a marker. Instead, it would be useful to add a negative marker. The Ponceau stain image should show the entire lane in order to represent the amount of loaded proteins: what is the molecular weight of the visible band? It appears to be UMOD, a protein characterized by high biological variability, which cannot be used as a “housekeeping”.
Moreover, please check the following inaccuracies and suggestions:
- Table 1: consider to add PRP, BOTOX and Tx to the legend;
- Figure 2: panel B, simplify the legend of the volcano plot; line 149 (Fig 2 caption), it is differential expressed proteins and not … expression proteins;
- Lines 170-173: rewrite the sentence;
- Figure 4 A: check the p-value in the p.adjust scale;
- Figure 5: check the figure title, it doesn’t seem right. Panel C: as validation of proteomic analysis, in addition to the WB images, there should also be quantifications with statistical analysis;
- 4.2 Urinary EV (uEV) isolation, line 269: please, check the cutoff, are you sure it is 100 Da?
- 4.3 Nanoparticle tracking analysis (NTA): please, describe the setting in more detail and add the type of laser (wavelength) used by the instrument;
- 4.9 Statistical analysis section, line 336: add “mean ± SEM”, because the quantitative data in figure 2 and 5 are expressed as mean ± SEM.
Kind regards
Author Response
Reviewer#1
Dear Authors,
I enjoyed reading your paper entitled “Proteomic Analysis of Urinary Extracellular Vesicles Supports 2 the “Toxic Urine Hypothesis” as a Vicious Cycle in Refractory 3 IC/BPS Pathogenesis” because I am convinced of the important role of urinary extracellular vesicles (uEV) in pathophysiological processes and their usefulness as an accessible source of biomarkers.
Responses:
We appreciate your positive feedback and acknowledgment of the scientific value of our study.
However, there are two main issues that need attention:
- As you yourselves admit, the low number of patients in the analysis make the study weak. I recommend emphasizing throughout the text that this is a pilot study with preliminary data.
Responses:
We agree with your comments. To improve the manuscript, we made revisions as follows:
- Title: Pilot Proteomic Analysis of Urinary Extracellular Vesicles Supports the Toxic Urine Hypothesis as a Vicious Cycle in Refractory ICBPS Pathogenesis (page 1, line 2-4)
- Abstract (in the third sentence): This pilot study examined uEV proteomics in refractory ICBPS cases…(page 1, line 22)
- Introduction (at the end of the last paragraph): This pilot case–control study explores proteomic changes in urinary EVs from a small cohort of refractory IC/BPS patients ….. effective interventions for IC/BPS. The data are intended as preliminary hypothesis-generating evidence to guide future larger-scale investigations. (page 2, line 77-85)
- Discussion:
- First sentence of the Discussion: This pilot proteomic analysis of uEVs from a small cohort of refractory IC/BPS patients revealed a paradoxical signature: upregulated homeostatic and regenerative proteins alongside persistent NF-κB activation. (page 8, line 227-229)
- A limitation related to the small number of samples is included as follows (in fifth paragraph): “Given the limited sample size, treatment heterogeneity, and absence of longitudinal sampling, the present work should be regarded as a pilot study providing preliminary hypothesis-generating data rather than definitive evidence. Future research should employ larger, standardized cohorts with serial sampling, correlate uEV profiles with cystoscopic and histopathological properties, and validate functions in cellular/animal models to confirm biomarkers and targets.”. (page 9, line 261-269)
- Conclusions:
- First sentence: “This pilot study provides an initial comprehensive proteomic characterization of urinary EVs from IC/BPS patients undergoing active treatment.” (page 11, line 384-385)
- Last 1–2 sentences: “Overall, the results of this pilot study emphasize that IC/BPS is not merely an inflammatory condition, but …. achieve sustained symptom relief and disease remission. Future studies with larger well-characterized cohorts are needed to validate these candidate biomarkers and mechanisms and to translate them into clinically actionable tools.” (page 12, line 395-403)
- A major issue concerns the choice of uEV isolation method, especially since the study is based on proteomics. Ultrafiltration is not suitable for proteomic analysis, since uEV purity is very low. In my experience, by ultrafiltration you can isolate uEV highly contaminated by albumin and Tamm Horsfall Protein/Uromodulin (UMOD). This aspect is not taken into consideration in the paper, nor is there any description of treatments that serve to reduce the influence of UMOD, which unfortunately has always complicated the proteomic analysis of uEV. Recent papers on uEVs that use ultrafiltration always associate it with Size-Exclusion Chromatography (SEC). Please comment on this question and explain the reason for choosing ultrafiltration. Moreover, it would be advisable to improve the characterization of EVs (Fig. 1E), showing the enrichment of markers in the vesicular fraction in comparison with at least total urinary proteins. Furthermore, since HSP70 is among the differential proteins, I would not use it as a marker. Instead, it would be useful to add a negative marker. The Ponceau stain image should show the entire lane in order to represent the amount of loaded proteins: what is the molecular weight of the visible band? It appears to be UMOD, a protein characterized by high biological variability, which cannot be used as a “housekeeping”.
Responses:
We appreciate the reviewer’s important comment regarding the impact of ultrafiltration on uEV purity and its implications for proteomic analysis. We fully agree that ultrafiltration alone cannot completely remove co‑isolated urinary proteins such as albumin and Tamm–Horsfall protein/uromodulin (UMOD), and that protocols combining ultrafiltration with additional purification steps (e.g., size‑exclusion chromatography) generally improve vesicle purity for in‑depth proteomics.
In this study, we deliberately chose ultrafiltration as the primary uEV isolation method to minimize sample loss from the limited urine volumes available, particularly from IC patients, and to ensure that sufficient material was retained for both EV characterization and discovery‑oriented proteomic profiling. Although our mass spectrometry data indeed show the presence of abundant contaminants such as albumin and uromodulin, we were still able to robustly detect and quantify a broad panel of EV‑associated proteins and to identify proteins with differential expression between healthy donors and IC patients. Importantly, our analyses focused on relative changes between groups processed using the same standardized workflow, rather than on absolute EV purity, which reduces the impact of co‑isolated proteins on the comparative conclusions.
To respond to this comment, we first remove HSP70 from the EV marker panel; EV characterization now focuses on canonical tetraspanin and ESCRT-associated markers (ALIX, TSG101, CD9) instead of HSP70. Second, a negative marker, calnexin, has been added, which is clearly detected in the HEK293T cell lysate but absent in both patient and healthy uEV samples, indicating minimal contamination by intracellular/endoplasmic reticulum proteins in the vesicular preparations. Third, the Ponceau S staining now shows two molecular weight ranges (75 to 60 kDa and 30 to 20 kDa) of original Figure 1E, thereby demonstrating the overall protein loading pattern and confirming that comparable total protein amounts were loaded for patient and healthy uEV samples; the full blot of Poneau S staining of calnexin data is also provided in supplement information. Finally, the revised figure also includes quantitative normalization values (e.g., ALIX, TSG101, CD9 intensities normalized to Ponceau S), which further illustrates enrichment of EV markers in the vesicular fraction relative to total lane protein and improves the rigor of EV characterization.
We also add a paragraph in the Discussion Section to discuss this limitation of our study as follow:
In recent years, the combination of ultrafiltration combination with size-exclusion chromatography (UF-SEC) has emerged as the most frequently employed protocol for uEV isolation in proteomic studies, owing to its superior removal of soluble contami-nants such as albumin and uromodulin [31]. In the present pilot study, we intentional-ly used ultrafiltration alone (with 100 kDa cutoff filters) to maximize EV recovery from the limited urine volumes available from our small clinical cohort (50 mL per subject). This choice preserved sufficient material for deep proteomic profiling; however, as ex-pected, it resulted in the co-isolation of abundant non-EV proteins, including albumin and uromodulin, which were identified in the LC-MS/MS data without differential ex-pression between groups. Western blotting confirmed the absence of cellular contami-nation (calnexin-negative; Figure 1E) and the presence of canonical EV markers (e.g., CD9, ALIX, TSG101) in both groups (Figure 1E). In the proteomic data, there were no statistical differences in the abundance of the canonical EV makers between IC/BPS pa-tients and healthy donors except for HSP70, which was increased in IC/BPS patients (Figure S1). Although more than 600 proteins were still identified in uEV samples from healthy donors or IC/BPS patients, the reduced dynamic range inherent to ultrafiltra-tion-only preparations likely masked lower-abundance candidates. For future valida-tion studies or larger cohorts, adopting UF-SEC or additional orthogonal purification steps (e.g., density-gradient centrifugation or uromodulin-reducing treatments) to achieve higher purity and higher confidence in low-abundance biomarkers should be considered [32]. (page 9, line 268-287 of clean version)
Moreover, please check the following inaccuracies and suggestions:
- Table 1: consider to add PRP, BOTOX and Tx to the legend;
Responses:
The following information has been added in the footnote as follows: “3 Tx, treatment; PRP, Platelet-Rich Plasma; BOTOX, OnabotulinumtoxinA”. (page 3, line 106 of clean version)
- Figure 2: panel B, simplify the legend of the volcano plot; line 149 (Fig 2 caption), it is differential expressed proteins and not … expression proteins;
Responses:
The legends of Figure 2B were revised as follows: “Volcano plot of the differential expressed proteins. The x-axis shows the log2 fold change, and the y-axis shows the -log10(p-value). Proteins with |logâ‚‚FC| > 1 and p < 0.05 are shown in red (sig-nificantly up- or downregulated). Gray dots represent non-significant proteins. Key proteins are labeled.” (page 5, line 158-161 of clean version)
- Lines 170-173: rewrite the sentence;
Responses:
We revised the descriptions of Figure 4B as follows: “The GSEA of GO C5 terms revealed the upregulation of metabolic processes including nucleotide metabolism, purine-containing compound metabolism, and mitochondrial organization in treated IC/BPS patients (Fig. 4B). On the other hand, pathways related to the innate immune defense showed downregulation, including defense responses to Gram-negative bacteria and fungi, antimicrobial humoral responses, and cell killing processes. In addition, external encapsulating structures also exhibited reduced enrichment (Fig. 4B). These patterns suggest metabolic activation accompanied by the suppression of innate immune responses in treated IC/BPS patients, potentially reflecting disease remission or therapeutic modulation of inflammatory pathways.” (page 6, line 179-188 of clean version)
- Figure 4 A: check the p-value in the p.adjust scale;
Responses:
We have confirmed that the p.adjust scale is correct. The descriptions of Figure 4A are now revised as follows: “Gene Set Enrichment Analysis (GSEA) using Hallmark gene sets showed enrichment trends toward mTORC1 signaling, MYC targets, glycolysis, estrogen response, fatty acid metabolism, and PI3K-AKT-mTOR signaling in treated IC/BPS patients, though these did not reach statistical significance after multiple testing correction (Fig. 4A). These patterns may suggest potential PRP-mediated metabolic and proliferative signaling during tissue regeneration.” (page 5-6, line 174-178 of clean version)
- Figure 5: check the figure title, it doesn’t seem right. Panel C: as validation of proteomic analysis, in addition to the WB images, there should also be quantifications with statistical analysis;
Responses:
These experiments aimed to validate the inflammation-induction effect of uEVs from refractory IC/BPS patient, and thus, we designed two experiments: (1) NF-kB reporter assay in HEK293T cells; (2) treating primary human bladder epithelial cells with uEVs from a healthy donor and a refractory IC/BPS patient and check the activation of NF-kB by western blot. Therefore, the title of Figure 5 has been revised a “Urinary extracellular vesicles from IC/BPS patients enhance NF-κB activity in HEK293T reporter cells and human bladder epithelial cells.”. (page 8, line 214-215 of clean version)
In addition, the quantifications of band intensities of panel C have been added.
- 2 Urinary EV (uEV) isolation, line 269: please, check the cutoff, are you sure it is 100 Da?
Responses:
We apologize for the mistake in the description of the Amicon filter's cutoff molecular weight. It should be 100 kDa. (page 10, line 308 of clean version)
- 3 Nanoparticle tracking analysis (NTA): please, describe the setting in more detail and add the type of laser (wavelength) used by the instrument;
Responses:
The description of the NTA method is now revised as follows:
The EV size and concentration were analyzed using NanoSight Pro (Malvern Panalytical Ltd., Malvern, UK) at the Medical and Pharmaceutical Industry Technology Development Center (New Taipei City, Taiwan). The instrument was equipped with a sCMOS camera, syringe-pump-driven flow cell for continuous sample delivery, and a 532 nm laser (maximum power <50 mW) for light-scatter detection. Samples were diluted in particle-free PBS to 20–100 particles/frame. Five 60 s videos were recorded per sample and analyzed using NTA software (Malvern Panalytical Ltd., Malvern, UK) with default settings, including finite track length adjustment and minimum track length enabled. (page 10, line 312-319 of clean version)
- 9 Statistical analysis section, line 336: add “mean ± SEM”, because the quantitative data in figure 2 and 5 are expressed as mean ± SEM.
Responses:
We thank the reviewer for this correction, which has been made to give “mean ±standard error of the mean (SEM)”. (page 11, line 378 of clean version)
Reviewer 2 Report
Comments and Suggestions for Authors
This study examined proteomic changes in urinary EVs between IC/BPS and healthy. The authors identified differential expressed proteins and enriched pathways, inferring potential and effective interventions for IC/BPS. Overall, this paper is well-organized and written. It is recommended that the paper be accepted after the authors address the following minor points:
- Need more discussion about the results. For example, what is the difference of the canonical EV markers in Figure 1E?
- There are no statistical analysis or labels in Figure 1F and 1G.
- What about the expression level of canonical EV markers in 1E in the mass spectrometry-based proteomic analysis? Do they show differential expression level between patients and healthy?
Author Response
Reviewer#2
This study examined proteomic changes in urinary EVs between IC/BPS and healthy. The authors identified differential expressed proteins and enriched pathways, inferring potential and effective interventions for IC/BPS. Overall, this paper is well-organized and written. It is recommended that the paper be accepted after the authors address the following minor points:
Responses: We appreciate your positive feedback and acknowledgment of the scientific value of our study.
- Need more discussion about the results. For example, what is the difference of the canonical EV markers in Figure 1E?
Responses:
We thank the comment from the reviewer. The descriptions have been revised as follows: “Western blot revealed the presence of canonical EV markers (ALIX, TSG101, CD9) in both in both groups, but not the negative marker calnexin (Fig. 1E). The expression levels of the EV markers were similar between the control group and the IC/BPS patients.”. (page 3, line 110-113 of clean version)
- There are no statistical analysis or labels in Figure 1F and 1G.
Responses:
There was no difference between groups. We added “n.s.” (not significant) to indicate that. (page 4 of clean version)
- What about the expression level of canonical EV markers in 1E in the mass spectrometry-based proteomic analysis? Do they show differential expression level between patients and healthy?
Responses:
After checking the proteomic data, there were no differences of canonical EV markers between groups, except to the significantly increased of HSP70 in IC/BPS patients. These data have been included as Figure S1 and being described in the last paragraph of “Discussion” section as follows:” Western blotting confirmed the absence of cellular contamination (calnexin-negative; Figure 1E) and the presence of canonical EV markers (e.g., CD9, ALIX, TSG101) in both groups (Figure 1E). In the proteomic data, there were no statistical differences in the abundance of the canonical EV makers between IC/BPS patients and healthy donors except for HSP70, which was increased in IC/BPS patients (Figure S1).”. (page 9, line 227-281 of clean version)

Round 2
Reviewer 1 Report
Comments and Suggestions for Authors
Dear Authors,
Thanks for responding to my comments and editing the manuscript where requested.
I would like to add a few more suggestions:
- when you write about EV protein markers, I suggest to use the adjective "typical" or "common" instead of "canonical";
- Fig. 1 E: don't report the quantification numbers under the western blot, but refere to figure S1;
- Fig. 5 C: thank you for showing the relative quantification, but the statistical analysis is missing. Please, add a new supplementary figure (S2) similar to S1; if the results are not significantly different due to the low number of samples, you can at least show a trend that will need to be confirmed on a larger cohort, as you wrote in the text.
Kind regards
Author Response
Reviewer#1
Thanks for responding to my comments and editing the manuscript where requested.
I would like to add a few more suggestions:
- when you write about EV protein markers, I suggest to use the adjective "typical" or "common" instead of "canonical";
Responses:
We thank the suggestion from the reviewer. In this revision, the “canonical” has been replaced by “common” through whole manuscript.
- Fig. 1 E: don't report the quantification numbers under the western blot, but refere to figure S1;
Responses:
We appreciate the reviewer's suggestion to avoid displaying quantification numbers under the Western blot in Figure 1E and instead refer to Figure S1. The Western blot in Figure 1E serves primarily as a qualitative quality control step in the study workflow: to confirm the successful isolation of uEVs by demonstrating the presence of canonical EV markers (ALIX, TSG101, CD9) and the absence of cellular contamination (calnexin-negative), with visually comparable marker expression between healthy controls and IC/BPS patients. This QC was performed prior to subjecting the same uEV samples to LC-MS/MS proteomic analysis.
In contrast, Figure S1 presents quantitative proteomic abundance data (from label-free quantification in the mass spectrometry dataset) for a broader panel of canonical EV markers, including HSP70/HSPA1A, which showed significant upregulation in patient uEVs.
The densitometric numbers originally displayed below the blots in Figure 1E were derived from semi-quantitative analysis of the immunoblot signals normalized to Ponceau S staining, supporting the textual statement that marker expression levels were similar between groups.
However, to align with common practices in EV research (as per MISEV guidelines, where Western blots for markers like CD9, ALIX, and TSG101 are typically presented qualitatively for presence), we have retained the blots without overlaid quantification numbers. The figure legend and main text have been revised as follows:
- Main text: “The expression levels of the EV markers were similar between the control group and the IC/BPS patients, supporting successful uEV isolation for subsequent proteomic analysis. Quantitative abundance of theses common EV markers by mass spectrometry was pro-vided in Fig. S1.” (page 3, line 113-115, in the clean version)
- Legends of Figure 1E: “….Ponceau S staining serves as the loading control. Cell lysate of HEK 293T cells was used as a control for calnexin detection. Quantitative proteomic abundance of these common EV markers is shown in Figure S1. The numbers below the blots indicate expression levels relative to healthy controls after normalization to Ponceau S signals.”. (page 4, line 125-126, in the clean version)
- Fig. 5 C: thank you for showing the relative quantification, but the statistical analysis is missing. Please, add a new supplementary figure (S2) similar to S1; if the results are not significantly different due to the low number of samples, you can at least show a trend that will need to be confirmed on a larger cohort, as you wrote in the text.
Responses:
We thank the reviewer for the positive feedback on including relative quantification in Figure 5C and for the valuable suggestion to add statistical analysis.
Figure 5C shows representative Western blots from functional validation experiments in primary human bladder epithelial cells (HBlEpC). Due to the limited yield of urinary EVs from individual urine samples (particularly from patients with low voided volumes), we were able to perform this experiment using uEVs from only one healthy donor and one IC/BPS patient, each tested in technical triplicate lanes to ensure reproducibility of the observed upregulation of 14-3-3ζ and phosphorylated NF-κB p65.
Given that this is a representative result from n=1 biological replicate per group, formal statistical comparison between groups is not appropriate.
We have therefore retained the relative densitometric quantification numbers below the bands in Figure 5C (normalized to GAPDH) to illustrate the observed trend in this representative experiment, as originally shown. To address the reviewer’s concern, we have revised the figure legend and the main text (Section 2.5) to explicitly state the limited sample size (n=1 per group) as follows:
“We further treated primary human bladder epithelial cells (HBlEpC) with the isolated uEVs. Due to limited uEV yield from individual samples, this functional validation was performed using uEVs from one healthy donor and one IC/BPS patient. Patient-derived uEVs upregulated 14-3-3ζ and phosphorylated NF-κB p65 (ser536) (Fig. 5C).” (page 7, line 209-231, in the clean version).
We also add some descriptions about this limitation in the Discussion section as follows:
“Additionally, functional validation of NF-κB activation and 14-3-3ζ upregulation in primary bladder epithelial cells was limited to uEVs from single representative healthy and IC/BPS samples due to constrained EV yields, precluding statistical analysis. Future studies with optimized EV isolation protocols or pooled samples should validate these trends in larger cohorts.” (page 9, line 270-274, in the clean version).